# Effects of Pine Pollen Polysaccharides and Sulfated Polysaccharides on Ulcerative Colitis and Gut Flora in Mice

**DOI:** 10.3390/polym15061414

**Published:** 2023-03-13

**Authors:** Yali Wang, Xiao Song, Zhanjiang Wang, Zhenxiang Li, Yue Geng

**Affiliations:** Key Laboratory of Food Nutrition and Safety of SDNU, Provincial Key Laboratory of Animal Resistant Biology, College of Life Science, Shandong Normal University, Jinan 250014, China

**Keywords:** pine pollen polysaccharides, ulcerative colitis, intestinal immunity, serum metabolomics, gut flora

## Abstract

Polysaccharides are important biological macromolecules in all organisms, and have recently been studied as therapeutic agents for ulcerative colitis (UC). However, the effects of *Pinus yunnanensis* pollen polysaccharides on ulcerative colitis remains unknown. In this study, dextran sodium sulfate (DSS) was used to induce the UC model to investigate the effects of *Pinus yunnanensis* pollen polysaccharides (PPM60) and sulfated polysaccharides (SPPM60) on UC. We evaluated the improvement of polysaccharides on UC by analyzing the levels of intestinal cytokines, serum metabolites and metabolic pathways, intestinal flora species diversity, and beneficial and harmful bacteria. The results show that purified PPM60 and its sulfated form SPPM60 effectively alleviated the disease progression of weight loss, colon shortening and intestinal injury in UC mice. On the intestinal immunity level, PPM60 and SPPM60 increased the levels of anti-inflammatory cytokines (IL-2, IL-10, and IL-13) and decreased the levels of proinflammatory cytokines (IL-1β, IL-6, and TNF-α). On the serum metabolism level, PPM60 and SPPM60 mainly regulated the abnormal serum metabolism of UC mice by regulating the energy-related and lipid-related metabolism pathways, respectively. On the intestinal flora level, PPM60 and SPPM60 reduced the abundance of harmful bacteria (such as *Akkermansia* and *Aerococcus*) and induced the abundance of beneficial bacteria (such as *lactobacillus*). In summary, this study is the first to evaluate the effects of PPM60 and SPPM60 on UC from the joint perspectives of intestinal immunity, serum metabolomics, and intestinal flora, which may provide an experimental basis for plant polysaccharides as an adjuvant clinical treatment of UC.

## 1. Introduction

Ulcerative colitis (UC) is typified by mucosal inflammation of the colon, which is one form of inflammatory bowel disease (IBD). The main clinical manifestations of UC are diarrhea, abdominal pain, bloody stools, and weight loss [1,2,3]. IBD is a chronic recurrent intestinal inflammatory disease. With its continuously increasing incidence rate, it has become a crucial public health burden globally [4]. At present, the main clinical drugs for the treatment of UC are aminosalicylate, corticosteroids, thiopurine, methotrexate, and anti-TNF-α drugs [5]. However, these pharmaceuticals lead to some side effects in patients over long-term administration, including nausea, vomiting, heartburn, diarrhea and headache [6]. Therefore, it is of great significance to explore natural products without side effects as adjuvant treatments for improving the treatment of UC.

Polysaccharides are important biological macromolecules in all organisms, and they are usually extracted from various plants, animals, fungi, bacteria, and algae. Because of their biodegradability, non-toxicity, and biocompatibility, polysaccharides have been studied as therapeutic agents for many chronic diseases over the past few decades [7]. In recent years, polysaccharides from natural resources have become the research focus of UC disease due to their good safety profile. These polysaccharides are usually related to the regulation of inflammatory cytokines, intestinal flora, immune system and the protection of intestinal mucosa in the treatment of UC. For example, *Ganoderma lucidum* polysaccharides (GLP) significantly inhibited the secretion of cytokines (e.g., TNF-α, IL-1β, IL-6, IL-17A, and IL-4) and maintained intestinal homeostasis in DSS-induced UC mice [8]. *Astragalus* polysaccharides (APS) protected the colon by enhancing levels of suppressor T-reg cells and reducing the secretion of IL-17 to ameliorate TNBS-induced colitis in rats [9]. The crude polysaccharides of Korean persimmon vinegar (KPV-0) significantly increased the secretion of intestinal juice and the immunoglobulin content A of feces [10]. *Hericium erinaceus* polysaccharides (HEP) significantly repaired injury of the intestinal mucosa and significantly increased the levels of sIgA, IFN-γ, and IL-4 to enhance intestinal mucosal immunity in young Muscovy ducks [11]. Therefore, we speculate that pine pollen polysaccharides may affect the intestinal immunity of UC mice.

Gut flora plays a key role in the metabolism, development, and function of the host immune system [12]. The active phase of IBD is accompanied by an imbalance in the gut flora [13] and decreased flora biodiversity. Furthermore, during this process, the abundance of Verrucomicrobia, Firmicutes, Bacteroidetes, Proteobacteria, and Actinobacteria was significantly changed [14,15,16]. For example, harmful bacteria such as *E. coli* were increased and beneficial bacteria such as *Alistipes*, *Roseburia*, and *Ruminococcus* were decreased [17]. Shao et al. found that polysaccharides of *Hericium erinaceus* mycelia improved the diversity of microorganisms and simulated the production of short-chain fatty acids (SCFAs), thereby restoring the disordered abundances of Firmicutes, Bacteroidetes, Proteobacteria, and Actinobacteria and alleviating the UC mice [14]. Both unfermented *Yupingfeng* polysaccharides and fermented *Yupingfeng* polysaccharides regulated the balance of gut flora and maintained the healthy intestinal barrier structure and function of weaned rex rabbits [18]. Extracellular polysaccharides (EPS1-1) from *Rhizopus nigricans* fermentation regulated the intestinal microflora of colorectal cancer model mice and increased the fecal content of SCFAs [19]. Therefore, we speculate that pine pollen polysaccharides may affect the gut flora of UC mice.

Recently, the sulfating of polysaccharides is of increasing concern. After artificially sulfated modification, sulfated natural polysaccharides usually exhibit higher biological activities, such as antiviral [20], antioxidant [21], immunomodulatory [22], and anti-aging properties [23]. It has been shown that sulfated modified *Astragalus* polysaccharides have stronger antiviral activity against the Fasciola virus when compared to unmodified *Astragalus* polysaccharides [24]. Pumpkin polysaccharides modified by sulfating showed enhanced ability to scavenge superoxide anions, which partially improved the antioxidant activity of the polysaccharides [25]. The sulfated polysaccharides and their sulfated derivatives attenuated the LPS-induced inflammatory response by inhibiting the phagocytosis of macrophages, NO production, and the release of IL-6 and IL-1β. In addition, the effect of sulfated *Ganoderma lucidum* polysaccharides was more pronounced, indicating that the sulfated modification improved the anti-inflammatory effect of *Ganoderma lucidum* polysaccharides [26]. Sulfated *Ganoderma lucidum* polysaccharides had a stronger ability to remove DPPH radicals than their natural counterparts. They also exhibited higher immunomodulatory activity by increasing macrophage phagocytosis and TNF-α production [27]. Nevertheless, the role of sulfated polysaccharides in UC requires validation through further study.

Our previous lab work confirmed that the sulfated polysaccharides of Masson pine pollen inhibited the proliferation of HepG2 cells [28], activated macrophages, and improved the body’s immune capability [29]. Therefore, we wondered whether sulfated *Pinus yunnanensis* pollen polysaccharides also lessened the progress of intestinal-related diseases. In this study, *yunnanensis* pine pollen polysaccharides were sulfated. The effects of *yunnanensis* pine pollen polysaccharides and sulfated polysaccharides on UC progression were then investigated at three levels: immunity, metabolomics and metagenomic, thus providing an empirical basis for the exploitation of polysaccharides as clinical drugs.

## 2. Materials and Methods

### 2.1. Extraction of Polysaccharides

Broken *Pinus yunnanensis* pollen was provided by the Yantai New Era Health Industry Company (broken rate > 95%). Broken *Pinus yunnanensis* pollen polysaccharides were extracted using the water extract–alcohol precipitation method. *Pinus yunnanensis* pollen and distilled water in a ratio of 1:14 were boiled for 4–5 h. Then, the supernatant was collected and concentrated to 10% of the original volume by rotary evaporation. Next, three-fold volume of 40%, 60% and 80% ethanol were added and mixed overnight at 4 °C for graded precipitation. The next day, white flocculent precipitate was collected. Considering the yield and biological activity, the polysaccharide precipitated by 60% ethanol concentration was selected for subsequent experiments and named as PPM60. Finally, proteins were removed using the trichloroacetic acid precipitation method [30]. 

### 2.2. Monosaccharide Composition

The monosaccharide composition of PPM60 was analyzed using 1100 high-performance liquid chromatography system (HPLC, Agilent, Santa Clara, CA, USA). Briefly, 5 mg of PPM60 was hydrolyzed with 0.25 mol/L H_2_SO4 at 100 °C for 4 h. After cooling, the mixture was completely neutralized with barium carbonate overnight. The supernatant was centrifuged. Next, the 15 µL PPM60 hydrolysate and the standard monosaccharide mixture were vacuum lyophilized. Then, 20 µL PMP (0.5 mol/L) and NaOH (0.3 mol/L) were added and reacted in a water bath at 70 °C for 30 min. 25 µL of hydrochloric acid solution (0.3 mol/L) was then used to neutralize. Finally, 0.5 mL of isoamyl acetate was conducted to the extracted hydrolysate. The HPLC parameters were as follows: Column, Diamonsil-Cl8 (5 µm, 150 × 4.6 mm); Column temperature, room temperature; Flow rate, 1.0 mL/min; Mobile phase, solvent A, acetonitrile; solvent B, 0.2 mol/L H_2_KPO_3_–0.1 mol/L NaOH, pH 6.8; Elution conditions, solvent A:solvent B = 17:83. Detection wavelength, 250 nm (4 nm band width); reference wavelength, 360 nm (4 nm band width); injection volume, 20 μL.

### 2.3. Preparation of Sulfated Polysaccharides

A sulfate derivative, named SPPM60, was obtained using chlorosulfonic acid pyridine. In a fume hood, 2 mL of pyridine was added to the pre-chilled mortar. Then, under ice bath conditions, 2 mL of chlorosulfonic acid was slowly added along the wall while stirring rapidly and continuously. The ice bath was withdrawn as soon as white solid appeared, and subsequently stirred until a yellow-white sulfonated reagent was produced. Later, the formamide polysaccharide solution (6 mL/100 mg) was slowly added and reacted at 45 °C for 4 h. Afterwards, the solution was neutralized with NaOH and transferred to a dialysis bag (Solarbio, Beijing, China) for purification. After dialysis until colorless, the liquid was concentrated using SB-1000 rotary evaporator (EYELA, Tokyo, Japan) and the concentrated liquid was freeze-dried by FDU-1200 freeze-dryer (EYELA, Tokyo, Japan). Finally, the barium sulfate turbidity method was conducted to assess sulfur content, and the degree of substitution. 

### 2.4. Infrared Spectroscopy Assay

The characteristic absorption peaks of PPM60 and SPPM60 were determined using iS 50 FT-IR spectrophotometer (Thermo, Waltham, MA, USA). PPM60 and SPPM60 were powdered and pressed into tablets. Then, the samples were measured in the frequency range of 4000–400 cm^−1^ [28]. 

### 2.5. Animals and DSS Colitis Model

SPF Male C57BL/6 mice, weighing 18–20 g (six weeks old), were purchased from Jinan Pengyue Laboratory Animal Breeding Co., Ltd. (Quality Certification of Laboratory Animals: SCXK (Lu) 20140007). This animal study was conducted in accordance with the ethical guidelines approved by the Committee for the Protection and Use of Shandong Normal University Animals (No. AEECSDNU2019042). Forty mice were randomly divided into four groups (n = 10): healthy control group (HC); model group (DSS); *Pinus yunnanensis* pollen polysaccharides group (PM); and sulfated polysaccharides group (SPM). UC mice were induced using 3% DSS (MP Biomedicals, Santa Ana, CA, USA) water solution, replaced with normal water after five days. The control group was given normal water (see Table 1). At the same time each day, mice in PM and SPM groups were treated with 200 mg/kg dose of PPM60 and SPPM60 by intragastrical administration (i.g), respectively. The HC and DSS groups were given equal volumes of drinking water in the same manner. In addition, PPM60 and SPPM60 treatment was used at the modeling stage simultaneously to conduct the prevention program. The mice were weighed, fed, and checked for conditions (e.g., activity, backward hair, etc.) daily. After seven days, the mice were euthanized by cervical dislocation after anesthesia, while biological materials such as blood, colon tissue, feces and cecum contents were collected for subsequent experimental analysis.

### 2.6. Clinical Disease Scores

According to a mentioned method [31], the disease activity index (DAI) of mice was calculated daily to evaluate the severity of the disease. Briefly, the animals were evaluated for weight loss (0–4), stool consistency (0–4), and blood in the stool (0–4), resulting in a maximum DAI score of 12.

### 2.7. Hematoxylin and Eosin Staining

The colon was fixed in 4% formaldehyde solution (Servicebio, Wuhan, China). After adequate fixation, colon tissue was embedded and pathological tissue sections were prepared. H&E staining was then performed to evaluate colon injury and inflammation. 

### 2.8. ELISA

The colon tissues were washed with pre-cooled physiological saline and dried with filter paper. Saline was then added at a ratio of 1:9. Next, the tissues were ground for 30 s and stopped for 30 s by a biological sample homogenizer (Aosheng Instrument, Hangzhou, China) three times, and then placed on ice for 5 min. After three cycles, the homogenate were centrifuged at 12,000 rpm/min at 4 °C for 20 min Finally, supernatants were collected to detect the expression of IL-1β, IL-6, TNF-α, IL-2, IL-10, IL-13, and sIgA by corresponding ELISA Kit (Multi Sciences, Hangzhou, China). In addition, the levels of serum CRP and D-lactic acid were determined by the CRP (Multi Sciences, Hangzhou, China) and D-lactic acid (Jiancheng Technology, Nanjing, China) kits according to the manufacturer’s instructions. The absorbance values were measured using Spectra Max Plus Microplate Reader (Molecular Devices, San Jose, USA) and the expression of each molecule was calculated according to the standard curve formula.

### 2.9. Sample Preparation and 1H-NMR Spectroscopy

Frozen serum samples were thawed to perform NMR analysis. Then, 500 µL of phosphate buffer solution (K_2_HPO_4_-NaH_2_PO_4_, 0.1 M, pH 7.4; 30% D_2_O containing 0.1% TSP, Sigma-Aldrich, St. Louis, MO, USA) was added to 100 µL of serum for nuclear magnetic resonance (NMR) detection. After centrifugation at 10,000 rpm for 10 min at 4 °C, approximately 550 µL of the clear supernatant was transferred into 5-mm NMR tubes (Wilmad-Lab Glass, Warminster, PA, USA) for sampling preparation.

All ^1^H-NMR spectra were obtained using a superconductor shielding FT NMR spectrometer equipped with ^13^C and ^1^H double-resonance optimization of a 5-mm CPTCI three-trans detector CryoProbeTM AVANCE 400 III (Bruker, Billerica, Germany) under the following conditions: 400.13 MHz proton resonance frequency, zg30 pulse sequence, 8012.8 Hz spectral width, number of scans was 256 at 298 K, 1 s relaxation delay, and 12 μs pulse length. Topspin 3.2 was used to process the spectral data.

### 2.10. 16S rDNA Amplicon Sequencing Analysis

Fecal (F) and cecum content (C) samples were freshly collected under aseptic conditions. The samples were stored at −80 °C until detection. The genomic DNA of the samples were extracted using the cetyltriethylammonium bromide method, and the purity and concentration of the DNA were detected via agarose gel electrophoresis. PCR amplification was carried out on the sequence with the upstream primer ′5-CCTACGGRRBGCASCAGKVRVGAAT-3′ and the downstream primer ′5-GGACTACNVGGGTWTCTAATCC-3′ of 16s V3 and V4 variable regions. A library was constructed using the Ion Plus Fragment Library Kit 48 rxns and was quantified using Qubit and Q-PCR. After qualification, the library was sequenced by Illumina MiSeq sequencing platform (GENEWIZ, Suzhou, China). 

The raw data were first spliced and filtered to obtain clean data. Operational taxonomic unit (OTU) clustering and species classification analysis were then performed based on valid data. Species annotations were made for the representative sequences of each OTU, and the corresponding species information and species-based abundance distributions were obtained. At the same time, the OTUs were analyzed for abundance, alpha-diversity, Venn plots, and petal plots to obtain species richness and uniformity information for the samples. Information regarding the common and unique OTUs between different samples or groups were also obtained. Statistical analysis methods, such as *t*-test and MetaStat, were used to test differences in the species composition and community structure of the grouped samples to further explore variations in the community structures between the grouped samples.

### 2.11. Statistical Analyses

Statistical analyses were performed using the paired Student’s *t*-test or multi-way analysis of variance. Statistical analyses were conducted using GraphPad Prism software (GraphPad, Inc., La Jolla, CA, USA). Differences were considered statistically significant when *p* < 0.05. All experiments were conducted independently at least three times.

## 3. Results

### 3.1. Monosaccharide Composition and Sulfating of PPM60

According to the results of HPLC, PPM60 is mainly composed of galactose, glucose, xylose, mannose, rhamnose, and an unknown monosaccharide with molar ratio as 12.830:10.449:29.693:1:1.415:1.426 (Figure 1a). The substitution degree of sulfuric acid was 1.45. IR spectroscopy showed that PPM60 had an absorption peak of O-H, with a stretching vibration at 3372.63 cm^−1^, and a stretching vibration of C-H appeared at 2929.58 cm^−1^. These are the characteristic absorption peaks of polysaccharides. SPPM60 had a characteristic absorption peak of S=O at 1224.56 cm^−1^ and a characteristic absorption peak of C-O-S at 831.45 cm^−1^. In addition, the absorption peak of the O-H stretching vibration of SPPM60 was weaker than PPM60, indicating that -OH was replaced by SO_4_^2−^ (Figure 1b). Furthermore, the substitution degree of sulfation was 1.45. These results demonstrated that we successfully obtained PPM60 and SPPM60.

### 3.2. Effects of PPM60 and SPPM60 on Disease Progression in UC Mice

By monitoring the body weight of the mice in each group, we found that the HC group displayed an overall increasing trend and were in a normal growth state. In contrast, the bodyweight of the DSS group showed a sharp downward trend on the fifth day, even exceeding 20% by the seventh day. The DAI scores of the DSS groups also increased sharply on the same day. While the PM and SPM groups also had a significantly reduced body weight and increased DAI score on day five, the overall trend was less than in the DSS group. (Figure 2a,b). In addition, the DSS mice began to develop loose and/or bloody stools, and the average length of colons was 4.7 ± 0.28 cm, which was significantly shorter compared to the HC group (6.7 ± 0.18 cm). Notably, PPM60 and SPPM60 treatment partially ameliorated the colon shortening phenomenon in PM (5.2 ± 0.09 cm) and SPM (5.6 ± 0.17 cm) mice, respectively (Figure 2c). Figure 2d shows the histologically stained colon tissue of mice from the four groups. The colon structure of the HC group was intact, without damage to the mucosal layer, and the epithelial cells were not shed. In contrast, the colon of mice in the DSS group had obviously damaged mucosal layers, with the mucosal epithelial cells missing (black arrows), and connective tissue hyperplasia (red arrows). Moreover, the damage extended to the submucosa, and there were more inflammatory cell infiltrations in the mucosa and submucosa, including lymphocytes and granulocytes (yellow arrows). In the PM group, the mucosal layer of the colon tissue was also damaged. Differently, the structure of the mucosal epithelium was intact, and there was a small amount of visible connective tissue hyperplasia, as well as a few visible inflammatory cell infiltrations into the mucosa and submucosa (red arrow). Otherwise, there were no other obvious abnormalities. In the SPM group, there was damage to the local mucosal layer of the colon tissue and a small amount of connective tissue hyperplasia (red arrow). The damage extended to the submucosa, and there was a small amount of inflammatory cell infiltration into the mucosa and submucosa. These data reveal that the UC mice model was successfully established by DSS, whereas progression of the disease was inhibited by the PPM60 and SPPM60 treatment.

### 3.3. Effects of PPM60 and SPPM60 on Intestinal Immunity in UC Mice 

Compared with mice in the HC group, the levels of IL-1β, IL-6, and TNF-α was obviously increased in the DSS group, indicating that there was inflammation in the colon tissue of the mice. The concentrations of IL-1β and IL-6 of mice in the PM and SPM groups displayed a downward trend, while the increase of TNF-α was not relieved (Figure 3a–c). In the DSS group, the concentrations of IL-2, IL-10, and IL-13 were decreased, which had opposite trends in the PM and SPM groups (Figure 3d–f). Serum D-lactic acid in the DSS group was slightly upregulated, which might be related to the hemolysis of samples in the HC group. On the contrary, D-lactic was markedly reduced in the PM and SPM groups (Figure 3g). Furthermore, CRP was clearly upregulated in the DSS group and was inhibited in the PM and SPM groups (Figure 3h). Moreover, there was a significant augmentation of sIgA in the DSS group, while SPPM60 notably reversed the increasing trend (Figure 3i).

These results demonstrate that PPM60 and SPPM60 had a positive effect on the intestinal inflammation of DSS-induced UC mice through alleviating intestinal inflammation. Furthermore, compared with PPM60, the effects of SPPM60 on intestinal immunity was slightly stronger.

### 3.4. Effects of PPM60 and SPPM60 on Serum Metabolomics in UC Mice

#### 3.4.1. Multivariate Statistical Analysis

According to the PCA model (Figure 4a), the HC group was clearly distinguished from the other three groups, but the DSS, PM and SPM groups were not clearly separated. However, in the model of PLS-DA and OPLS-DA (Figure 4b,c), the four groups were clearly separated, and the validation model of OPLS-DA showed that the model had not been overfitted (Figure 4d). We then compared the four groups of mice randomly in pairs; the HC group and DSS group were clearly distinct, and the OPLS-DA model test also showed that the results were relatively reliable (Appendix A). The DSS group and the PM group showed good separation in all three analysis methods (Appendix A). Although the DSS group and SPM group were not very distinct in the PCA analysis, the separation between these two groups in the PLS-DA and OPLS-DA was more obvious. The models were found to be more reliable (Appendix A).

#### 3.4.2. Qualitative Analyses of Serum Differentially Expressed Metabolites and Changes in Metabolic Pathways

To determine the differential expression of metabolites, we ranked the VIP values obtained by multivariate statistical analysis from high to low, and the chemical shift of VIP ≥ 2 was used to characterize the metabolites. Compared with the HC group, eight possible differentially expressed metabolites were identified in the DSS group (Table 2): N-acetyl-L-alanine, acetic acid, L-fucose, lactic acid, taurine, betaine, acetylcholine, and allose. Compared with the DSS group, seven potential differentially expressed metabolites were found in the PM group (Table 3): N-acetyl-L-alanine, acetic acid, lactic acid, fructose-6-phosphate, allose, D-xylose, and L-carnitine. There was possible differential expression of seven metabolites in the SPM group compared with the DSS group (Table 4): betaine, glyceryl phosphate, L-serine, D-xylose, acetylcholine, lactic acid, and allose.

The above-mentioned possible differentially expressed metabolites were entered into MetPA for analysis. As shown in Figure 5a, compared with the HC group, the main changed metabolic pathways of the DSS group mice included pyruvate metabolism, glycolysis/gluconeogenesis, taurine and hypotaurine metabolism, fructose and mannose metabolism, glycine, serine, and threonine metabolism, and primary bile acid biosynthesis. Signal-pathway enrichment analysis of the DSS group (Figure 5b) showed that the changes of the metabolic pathways mainly occurred in pyruvate metabolism, taurine and hypotaurine metabolism, ethanol degradation, betaine metabolism, phospholipid biosynthesis, fructose and mannose degradation, and other metabolic pathways. Compared with the DSS group, the PM group mainly exhibited the following abnormal metabolic pathways: pyruvate metabolism, glycolysis/gluconeogenesis, starch and sucrose metabolism, mutual conversion of pentose, and glucuronate interconversions (Figure 5c). A summary chart of the PM metabolic enrichment pathways mainly focused on amino sugar metabolism, gluconeogenesis, pyruvate metabolism, β-oxidation of long-chain fatty acids, ethanol degradation, carnitine synthesis, and other metabolic pathways (Figure 5d). Compared with the DSS group, the metabolic pathways of the SPM group were mainly associated with glycine, serine and threonine metabolism, glycerophospholipid metabolism, glycerolipid metabolism, pentose and glucuronic acid interconversion, glucuronate interconversions, glyoxylate and dicarboxylate metabolism, and aminoacyl-tRNA biosynthesis (Figure 5e). The enrichment analysis chart (Figure 5f) mainly involved phospholipid biosynthesis, methionine metabolism, glycine and serine metabolism, homocysteine degradation, de novo triacylglycerol biosynthesis, glycerol phosphate shuttle, cardiolipin biosynthesis and phosphatidylethanolamine biosynthesis, and other metabolic pathways.

### 3.5. Effects of PPM60 and SPPM60 on Gut Flora in UC Mice

Rarefaction curves are used to reflect the depth of sequencing (Figure 6a). Rank abundances were performed to reflect the richness and uniformity of the species in the sample (Figure 6b), and species accumulation boxplots (Figure 6c) were used to reflect the rate of new species appearance under continuous sampling. The curve and box plots gradually flattened with an increasing number of species, indicating sufficient sampling and the uniform distribution of species. In addition, increasing the data generated fewer new species. This indicated that the depth of sample sequencing gradually became reasonable to analyze the data.

For the contents of the mice cecum, the numbers of OTUs in HC, DSS, PM and SPM were 1117, 958, 1091, and 978, respectively (Figure 6d). In the mice feces, there were 1019, 906, 1003, and 1003 OTUs in the HC, DSS, PM and SPM groups, respectively (Figure 6e). Compared with the control HC group, there was a decreased number of OTUs in both the cecum contents and the feces of the DSS group mice. In addition, the species diversity was partially recovered by PPM60 and SPPM60 treatments.

Figure 6g shows the relative abundance of flora in the feces and cecum contents at the phylum level. Bacteroidetes and Firmicutes were the main gut flora, followed by Proteobacteria and Verrucomicrobia. Figure 6f shows the relative abundance of flora in the feces and cecum of mice in the four groups at the genus level. In the mice cecum contents, the most abundant bacteria of the HC group were *Bacteroides*, *Ileibacterium*, *Lactobacillus*, and *Odoribacter*. In the DSS group, the abundances of *Bacteroides* and *Akkermansia* were significantly elevated, while the abundance of *Ileibacterium* was significantly decreased. Instead, the abundance of *Akkermansia* was decreased in the PM and SPM groups. In addition, the abundance of *Ileibacterium* in the SPM group was obviously recovered. In the feces of mice, *Bacteroides*, *Ileibacterium*, and *Lactobacillus* were the dominant genera of the HC group. Compared with the HC group, the abundances of *Bacteroides*, *Helicobacter*, *Akkermansia*, *Odoribacter*, and *Turicibacter* were increased and the abundances of *Ileibacterium* and *Lactobacillus* were decreased in the DSS group. In the PM group, *Helicobacter*, *Lactobacillus*, and *Akkermansia* showed a recovery trend. In the SPM group, there was a regulatory effect on *Ileibacterium*, *Akkermansia*, *Odoribacter*, and *Turicibacter*.

Figure 7 shows the species with significant differences at the genus level. In the mice feces and cecum contents, there were significantly different abundances of *Aeroococcus*, *Akkermansia*, *Parasuttetella*, *Enterorhabolus*, *Parvibacter*, *Turicibacter*, *Lactobacillus*, *Romboutsia*, and *Arthromitus* among the groups. In the feces of mice of the DSS group, all the abundances of *Aerococcus*, *Akkermansia*, *Parasuttetella*, *Parvibacter*, *Turicibacter*, and *Romboutsia* were increased, while the abundances of *Enterorhabolus*, *Lactobacillus* and *Arthromitus* were decreased. Compared with the DSS group, the relative abundances of *Aerococcus*, *Akkermansia*, and *Parvibacter* were re-inhibited, while *Enterorhabolus*, *Lactobacillus*, and *Arthromitus* were re-promoted in the PM group. In the SPM group, *Aeroococcus*, *Akkermansia*, *Parvibacter*, *Turicibacter*, and *Romboutsia* were reduced and *Enterorhabolus* and *Lactobacillus* were elevated. Therefore, PPM60 and SPPM60 partially reversed the disorder of the gut flora and regulated different flora.

## 4. Discussion

Although UC is not directly life-threatening, it is often debilitating and can lead to dangerous complications. Therefore, it is important to explore the treatment of UC. *Pinus yunnanensis* Franch. is an evergreen coniferous tree of Pinaceae. It is distributed at altitudes between 400 m and 3100 m on mountains, river basins, dry and sunny slopes in Southwestern China, and is one of the main forest tree species [32]. Pine pollen is the male spore of *Pinus yunnanensis* Franch. According to the Chinese ancient medical code “Compendium of Materia Medica”, pine pollen has the functions of moistening the heart and lungs, benefiting qi, eliminating wind and stopping bleeding. As a traditional Chinese medicine and health food, it has a good relieving effect on fatigue, colds, prostate, anemia, diabetes, high blood pressure, asthma, etc. [33]. These beneficial effects are attributed to the various chemical constituents, including nucleic acids, enzymes and coenzymes, proteins, fats, acids, phospholipids, monosaccharides, polysaccharides, flavonoids, vitamins, etc. [34]. Thus, pine pollen has received a lot of attention in the food, biochemical and medical fields due to its high values for utilization. 

Sulfated polysaccharides are a group of polysaccharide derivatives with a complex steric structure and rich biological activity, with highly anticoagulant, antioxidant, antiproliferative, immune system modulating and antitumor effects [35]. However, natural sulfated polysaccharides are relatively scarce. Polysaccharides become acidic polysaccharides containing sulfate groups after sulfating, which can easily bind to specific structural domains of proteins, thus changing their conformation and affecting their biological activity [36]. In this study, a boiling alcohol precipitation method was conducted to extract. Previous experimental results illustrated that the content and activity of crude polysaccharide precipitated by 60% ethanol (PPM60) was high, hence we selected PPM60 for sulfating modification and the production was named SPPM60. We revealed that PPM60 was mainly composed of galactose, glucose, xylose, mannose, rhamnose, and an unknown monosaccharide. The sulfate substitution degree of SPPM60 was 1.45. In addition, SPPM60 also had the characteristic absorption peaks of S=O and C-O-S, and the characteristic absorption peaks of PPM60 were relatively reduced, indicating that the polysaccharide was successfully sulfated. Subsequently, because of its simplicity, reliability and good reproducibility, 3% DSS was used to induce acute UC in mice. The data shows that, compared with the HC group mice, the mice in the DSS group began to show a series of symptoms, including weight loss, increased DAI index, significantly shortened colon length, serious mucosal tissue damage, and a large number of inflammatory cell infiltrations, indicating that we successfully induced UC lesions in mic. Notably, PPM60 and SPPM60 had a certain effect on UC mice to improve the development and pathological changes. Specifically, colon shortening and injury was repaired. 

Serum CRP is a sensitive indicator of various infections and non-infectious inflammation in the body. It is a non-specific marker of systemic inflammation and one of the commonly used clinical indicators [37]. CRP is involved in the regulation of inflammatory response and is often used to reflect the inflammatory levels of IBD patients [38,39]. One of the pathophysiological characteristics of UC is a persistent intestinal inflammatory response, which can lead to chronic inflammation and tissue damage [40]. Studies have shown that the mRNA and proteins expressions of proinflammatory cytokines, including IL-1β, IL-6, and TNF-α, are increased in UC [41], while the levels of anti-inflammatory cytokines, including IL-4, IL-10 and IL-13, are decreased [42]. The anaerobic fermentation of carbohydrates by flora in the intestine produces D-lactic acid. Mammals only produce a small amount of endogenous D-lactic acid. Therefore, the increase of serum D-lactic acid may be a by-product of anaerobic fermentation by specific bacteria [43]. Clinical studies report that serum D-lactic acid of IBD patients is significantly higher than of control individuals, indicating that the intestinal mucosa of patients with acute IBD maybe damaged [44]. In addition, SIgA is also important for maintaining intestinal homeostasis [45]. Our results reveal that pro-inflammatory cytokines (IL-1β, IL-6, TNF-α), CRP, and serum D-lactic acid were significantly increased, while anti-inflammatory cytokine levels (IL-2, IL-10, and IL-13) were significantly reduced in the UC model group. However, after intervention with PPM60 and SPPM60, the trends of these cytokines, serum CRP and D-lactic acid were reversed. These results indicate that PPM60 and SPPM60 recovered the intestinal barrier by reducing inflammation in UC mice.

Non-targeted metabolomics is commonly used to detect changes of metabolites. This method can reveal disease-related metabolic disorders by measuring the changes of multiple metabolites in biological samples such as blood [46]. NMR is one of the main analytical techniques used in metabolomics. It has the advantages of simple sample preparation, good reproducibility, no sample damage and sample recycling [47]. IBD patients have reduced fatty acids, increased acylcarnitine levels, enhanced mitochondrial β-oxidation [48], and increased serum N-acetyl compounds during the active phase of the disease [49], indicating that they have a higher energy requirement for recruiting immune cells to fight inflammation [48]. In addition, the levels of glycolic acid, L-isoleucine, symmetrical dimethylarginine, serine, phosphoric acid ethanolamine, proline, and hexanoyl carnitine are upregulated in the serum of pediatric IBD patients [50].

Compared with the HC group, serum metabolite D-lactic acid levels were increased in the DSS group. On the contrary, D-lactic acid levels in the PM and SPM groups were decreased. These changes indicate that PPM60 and SPPM60 had some reparative effects on the damaged intestinal barrier. Furthermore, after PPM60 intervention, the levels of fructose-6-phosphate, allose and D-xylose were upregulated, whereas the levels of N-acetyl-L-alanine, acetic acid, and L-carnitine were downregulated compared with the DSS group. The pathway analysis also shows that these metabolites were mainly involved in pyruvate metabolism, glycolysis/gluconeogenesis pathways, pentose and glucuronate interconversions. After SPPM60 intervention, betaine, glyceryl phosphate, D-xylose, L-serine and acetylcholine were upregulated. These were involved in glycine, serine, and threonine metabolism, glycerophospholipid metabolism, glyoxylate and dicarboxylate metabolism, and aminoacyl tRNA biosynthesis pathways. Overall, these data illustrate that PPM60 and SPPM60 regulated and repaired the imbalance of serum energy and lipid metabolisms, thereby preventing the disease progression of UC mice. Differently, PPM60 mainly regulated the metabolic pathways related to energy metabolism, while SPPM60 mostly regulated the metabolic pathways related to lipid metabolism.

IBD is a series of complex diseases that cause chronic inflammation of the gastrointestinal tract. Compared with healthy individuals, the microbial diversity of IBD patients is greatly reduced [51]. In the small intestine and colon, segmented filamentous bacteria (SFB) can increase the number of host T-reg cells and regulate the differentiation of Th17 cells thereby promoting the production of IL-22-expressing CD4^+^ T cells [52]. *Aerococcus* is a harmful bacterium, which can cause diseases such as urinary tract infection (UTI), bloodstream infection (BSI), and endocarditis to spinal infection [53]. *Parasutterella* is associated with chronic intestinal inflammation, which is significantly increased in patients with irritable bowel syndrome (IBS) and IBS mice [54]. *Turicibacter* is a type of harmful Erysipelothrichaceae bacterium and is related to the development of Tsumura Suzuki obesity and diabetes [55]. *Lactobacillus*, a beneficial bacterium of the phylum Firmicutes, has antibacterial activity in vitro. It can be used as a functional food to effectively fight a variety of diseases [56,57,58]. Our experimental data showed that PPM60 and SPPM60 increased the abundance of beneficial bacteria, such as *Lactobacillus* and segmented filamentous bacteria (SFB). Instead, the abundance of harmful bacteria, such as *Aerococcus* and *Turicibacter* was reduced. By regulating the imbalance of gut flora, PPM60 and SPPM60 relieved UC progression.

In addition, we also found a significantly increased abundance of *Akkermansia* in DSS mice. *Akkermansia* is a mucin-degrading beneficial bacterium of the phylum Verrucomicrobia, and mucin is the only carbon source of *Akkermansia*. The presence of *Akkermansia* is associated with healthy intestines and is negatively correlated with several disease states [59]. *Akkermansia* can adhere to intestinal epithelial cells in vitro, which enhances the integrity of the intestinal cell monolayer and resists the impact of the damaged intestinal barrier [60]. The abundance of *Akkermansia* is reduced in obese and type 2 diabetic patients, and this reduction is related to poorer intestinal health and impaired metabolic status [61]. Studies have shown that, after DSS treatment, the abundance of *Akkermansia* significantly increases. Because *Akkermansia* specifically degrades mucin, the changes of this bacteria may be related to the amount of mucin in the intestine. *Akkermansia* can not only degrade mucin but can also stimulate mucin synthesis via an autocatalytic process [62]. Lipopolysaccharides (LPS) derived from gram-negative bacteria are the main causes of inflammation. Studies have found that the number of *Enterobacteriaceae* and *Akkermania* on the colonic mucosa were increased during the induction of colitis [63]. Nagalingam et al. found that, in the cecum of DSS-induced colitis mice, the abundance of Verrucomicrobia was increased, which may be related to its ability of metabolize sulfur and degrade mucin [64]. Our data show that *Akkermansia* abundance was significantly increased in the DSS group, while it was decreased in the PM and SPM groups. We analyzed that there were two reasons for this phenomenon: the first is the ability of *Akkermansia* to degrade and synthesize mucin, and the second is that *Akkermansia* is a gram-negative bacterium related to inflammation.

## 5. Conclusions

Both PPM60 and SPPM60 had ameliorative effects on DSS-induced UC, specifically in improving colon shortening, repairing intestinal injury, reducing intestinal inflammation, regulating serum metabolism, and regulating the balance of intestinal flora. Differently, PPM60 was more effective in inhibiting pro-inflammatory cytokines and mainly regulated metabolic pathways related to energy metabolism. Furthermore, PPM60 downregulated the abundance of harmful bacteria (*Akkermansia* and *Aerococcus*) and upregulated beneficial bacteria (*Lactobacillus* and *Arthromitus*). Conversely, SPPM60 tended to promote anti-inflammatory cytokines and mainly regulated metabolic pathways related to lipid metabolism. Moreover, SPPM60 downregulated the abundance of *Akkermansia*, *Aerococcus*, and *Turicibacter*, and upregulated *Lactobacillus*. To summarize, SPPM60 had a better effect on intestinal immunity than PPM60, while the recovery effect of intestinal flora species diversity was weaker than PPM60. However, this research could not completely determine which effect was better between PPM60 and SPPM60, and thus further research is still required.

## Figures and Tables

**Figure 1 polymers-15-01414-f001:**
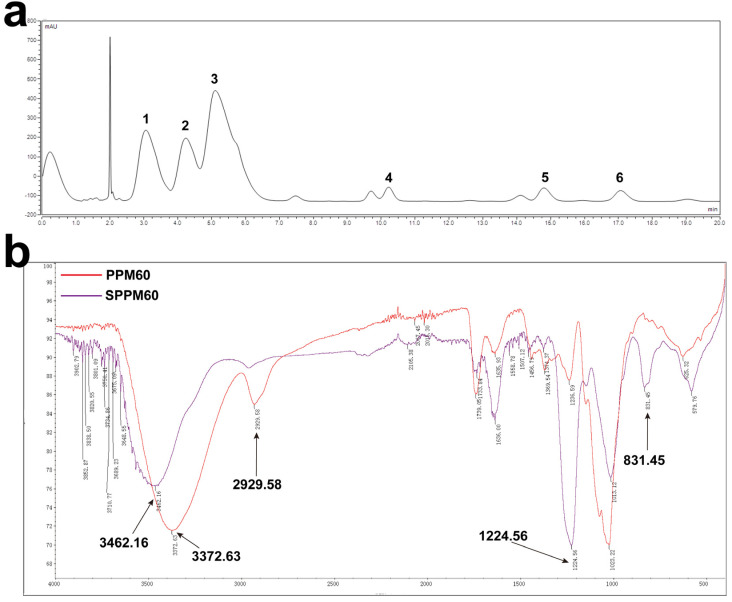
Extraction and sulfation of polysaccharides. (**a**) HPLC spectrum of PPM60; (**b**) infrared spectrum of PPM60 and SPPM60 sample. The numbers in the figure represent as follows: 1, galactose; 2, glucose; 3, xylose; 4, mannose; 5, rhamnose; 6, unknown.

**Figure 2 polymers-15-01414-f002:**
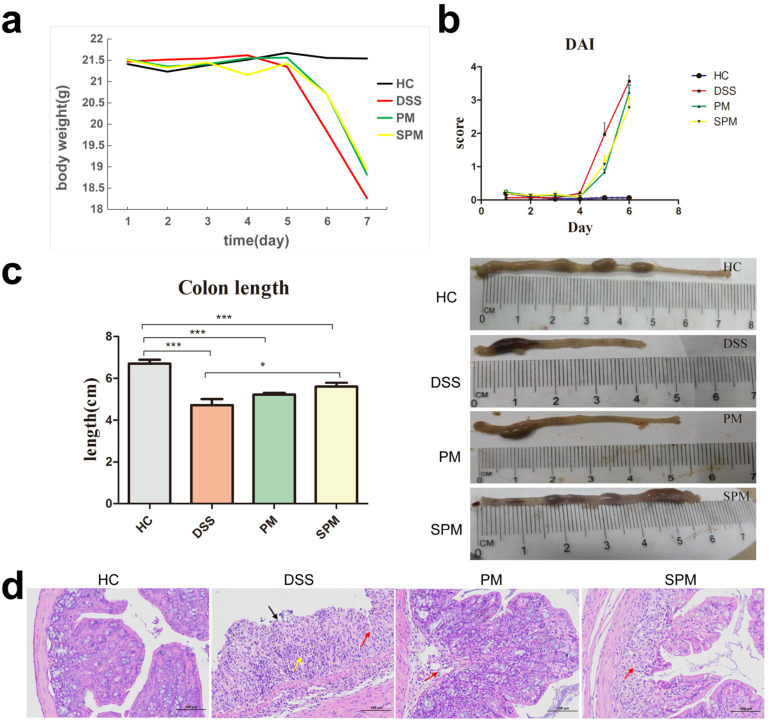
The disease progression of UC in mice. (**a**) Bodyweight change curve; (**b**) DAI score; (**c**) Colon length; (**d**) Colon tissue section (magnification, 200). DAI, Disease activity index; HC, healthy control group; DSS, model group; PM, *Pinus yunnanensis* pollen polysaccharides group; SPM, sulfated polysaccharides group. * *p* < 0.05, *** *p* < 0.001.

**Figure 3 polymers-15-01414-f003:**
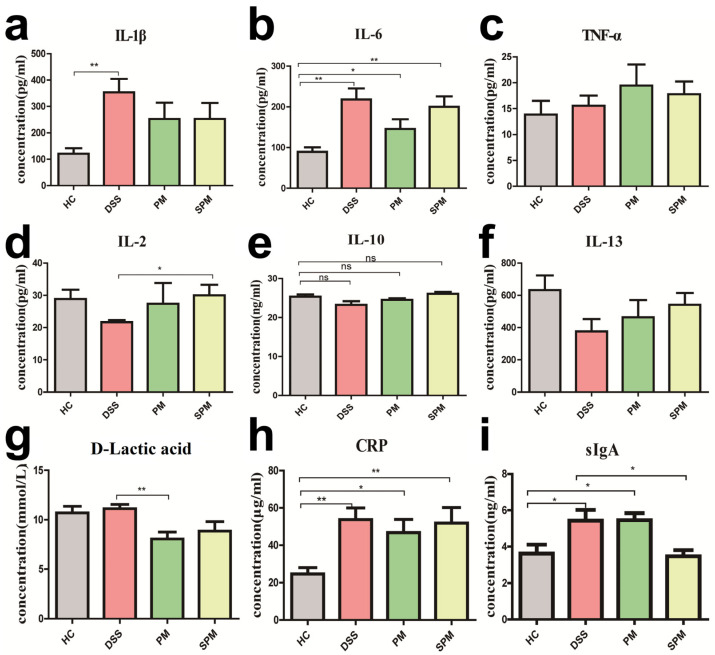
Expression of cytokines of colon tissues or serum in mice. (**a**) IL-1β; (**b**) IL-6; (**c**) TNF-α; (**d**) IL-2; (**e**) IL-10; (**f**) IL-13; (**g**) D-lactic acid; (**h**) CRP; (**i**) sIgA. The levels of IL-1β, IL-6, TNF-α, IL-2, IL-10, IL-13 and sIgA were measured in colon tissues. The levels of D-lactic acid and CRP were evaluated in serum. * *p* < 0.05, ** *p* < 0.01, ns: no significant difference.

**Figure 4 polymers-15-01414-f004:**
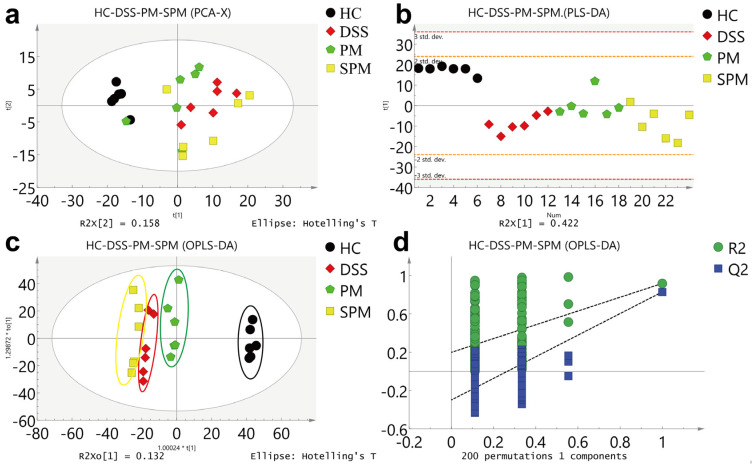
Multivariate statistical analyses of ^1^H NMR spectra of the HC, DSS, PM, and SPM groups. (**a**) PCA score map; (**b**) PLS score map; (**c**) OPLS score map; (**d**) The validation model of OPLS-DA.

**Figure 5 polymers-15-01414-f005:**
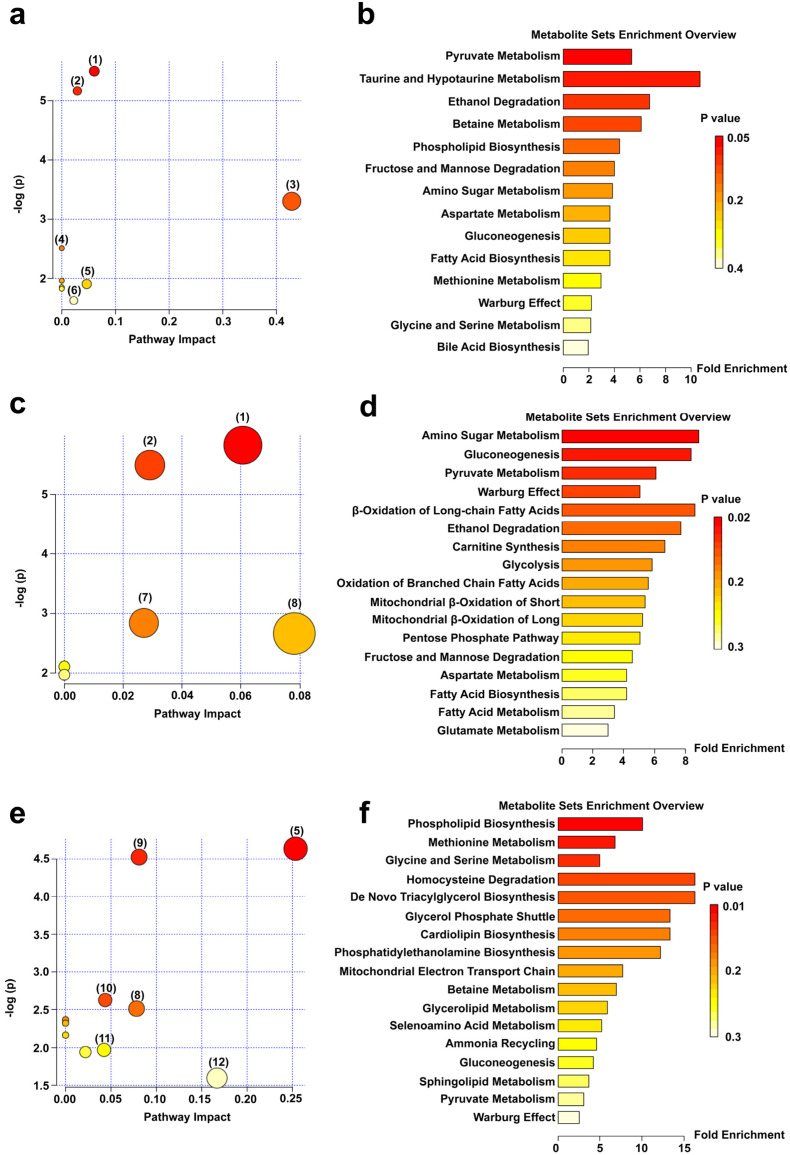
Metabolic pathway maps and enrichment overview maps of serum differentially expressed metabolites in mice. (**a**) Metabolic pathways and (**b**) Overview of metabolite enrichment in HC vs. DSS group; (**c**) Metabolic pathways and (**d**) Overview of metabolite enrichment in DSS vs. PM groups; (**e**) Metabolic pathways and (**f**) Overview of metabolite enrichment in DSS vs. SPM groups. The numbers in the metabolic pathway maps represent as follows: (1) Pyruvate metabolism; (2) Glycolysis/gluconeogenesis; (3) Taurine and hypotaurine metabolism; (4) Fructose and mannose metabolism; (5) Glycine, serine, and threonine metabolism; (6) Primary bile acid biosynthesis; (7) Starch and sucrose metabolism; (8) Pentose and glucuronate interconversions; (9) Glycerophospholipid metabolism; (10) Glycerolipid metabolism; (11) Glyoxylate and dicarboxylate metabolism; (12) Aminoacyl-tRNA biosynthesis.

**Figure 6 polymers-15-01414-f006:**
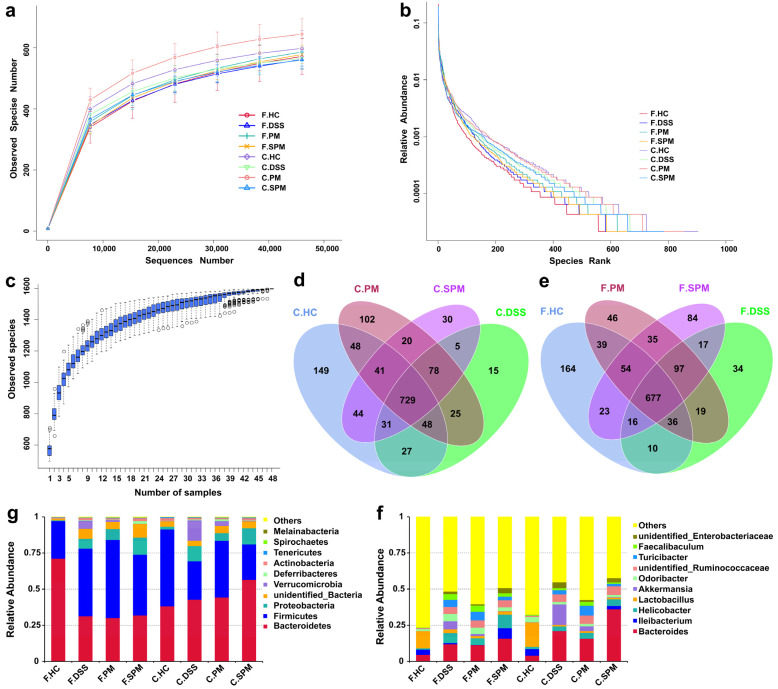
Sample complexity analysis and species distribution of the gut flora in mice. (**a**) Rarefaction curve; (**b**) Rank abundance curve; (**c**) Species accumulation boxplot; (**d**) OTUs Venn diagram of cecum contents; (**e**) OTUs Venn diagram of feces; (**g**) Relative abundance of species at the phylum level; (**f**) Relative abundance of species at the genus level. F, Feces; C, Cecum contents.

**Figure 7 polymers-15-01414-f007:**
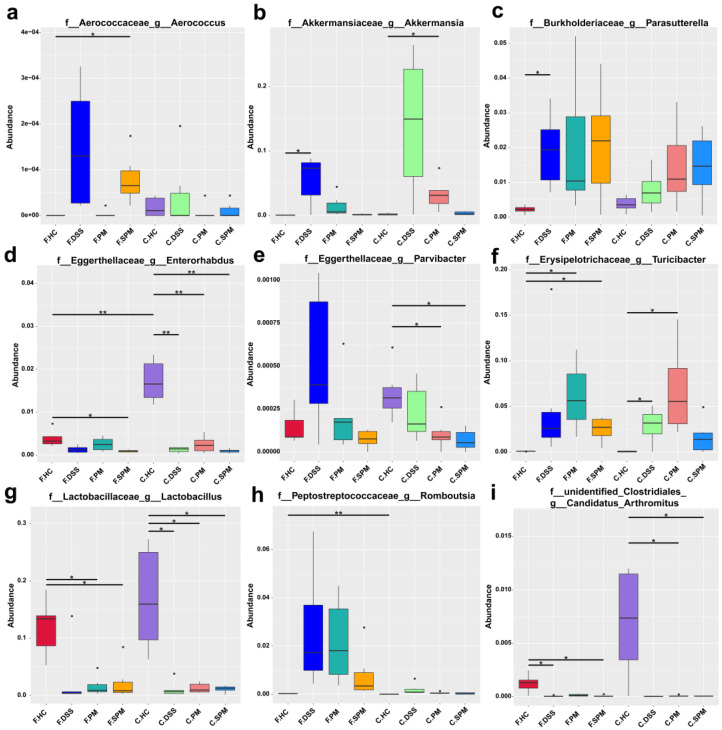
Significant differences in gut flora species abundances at the genus level. (**a**) *Aerococcus*; (**b**) *Akkermansia*; (**c**) *Parasutterella*; (**d**) *Enterorhabolus*; (**e**) *Parvibacter*; (**f**) *Turicibacter*; (**g**) *Lactobacillus*; (**h**) *Romboutsia*; (**i**) *Arthromitus.* * *p* < 0.05, ** *p* < 0.01.

**Table 1 polymers-15-01414-t001:** Illustration of the experimental treatment of mouse ulcerative colitis models.

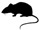 C57BL/6 male, 6 weeks
	Day 1	Day 5	Day 7 (euthanasia)

HC group	water (free drinking) + water (i.g)	water (free drinking) + water (i.g)	n = 10
DSS group	3%DSS (free drinking)+ water (i.g)	water (free drinking) + water (i.g)	n = 10
PM group	3%DSS (free drinking) +200 mg/kg PPM60 (i.g)	water (free drinking)+ 200 mg/kg PPM60 (i.g)	n = 10
SPM group	3%DSS (free drinking) +200 mg/kg SPPM60 (i.g)	water (free drinking) + 200 mg/kg SPPM60 (i.g)	n = 10

Note: i.g: intragastrical administration.

**Table 2 polymers-15-01414-t002:** Differently expressed metabolites in the DSS group compared with the HC group.

Name	Shifts	VIP	Trend
N-Acetyl-L-alanine	1.32 (d), 4.11 (t)	3.29	↑
Acetic acid	1.25 (t), 3.28 (s), 4.17 (q)	3.09	↑
L-Fucose	3.88 (dd), 3.79 (m), 3.64 (dd), 3.45 (dd), 1.25 (d)	3.22	↑
Lactic acid	4.1 (q), 1.32 (d)	2.42	↑
Taurine	3.25 (t), 3.42 (t)	2.76	↑
Betaine	3.89 (s), 3.25 (s)	3.10	↑
Acetylcholine	3.75 (t), 3.23 (s)	2.38	↑
Allose	3.41 (dd), 3.69 (dd), 3.78 (m), 3.87 (dd)	2.92	↓

Notes: s, singlet; d, doublet; t, triple peak; q, quartet; dd, doublet doublet; m, multiplet.

**Table 3 polymers-15-01414-t003:** Differently expressed metabolites in the PM group compared with the DSS group.

Name	Shifts	VIP	Trend
N-Acetyl-L-alanine	1.32 (d), 4.11 (t)	2.73	↓
Acetic acid	1.25 (t), 4.17 (q)	2.43	↓
Lactic acid	4.1 (q), 1.32 (d)	2.73	↓
Fructose-6-phosphate	3.92 (m), 3.65 (m), 3.55 (m)	3.26	↑
Allose	3.41 (dd), 3.63 (dd), 3.69 (dd), 3.87 (dd)	2.92	↑
D-xylose	3.21 (dd), 3.31 (dd), 3.42 (t), 3.51 (dd), 3.63 (m)	2.38	↑
L-carnitine	3.419 (s), 3.215 (s), 2.425 (s)	2.38	↓

Notes: s, singlet; d, doublet; t, triple peak; q, quartet; dd, doublet doublet; m, multiplet.

**Table 4 polymers-15-01414-t004:** Differently expressed metabolites in the SPM group compared with the DSS group.

Name	Shifts	VIP	Trend
Betaine	3.89 (s), 3.25 (s)	2.99	↑
Glyceryl phosphate	3.67 (dd), 3.82 (m)	2.24	↑
L-serine	3.832 (dd), 3.958 (m)	2.09	↑
D-xylose	3.21 (dd), 3.31 (dd), 3.42 (t), 3.51 (dd), 3.63 (m)	2.34	↑
Acetylcholine	3.75 (t), 3.23 (s), 2.15 (s)	2.43	↑
Lactic acid	4.1 (q), 1.32 (d)	3.00	↓
Allose	3.41 (dd), 3.63 (dd), 3.69 (dd), 3.78 (m)	2.33	↓

Notes: s, singlet; d, doublet; t, triple peak; q, quartet; dd, doublet doublet; m, multiplet.

## Data Availability

All data analyzed in this study were included in this article. Data sharing was not applicable to this article.

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
