# Peer review of "Effects of Pine Pollen Polysaccharides and Sulfated Polysaccharides on Ulcerative Colitis and Gut Flora in Mice"

_polymers, 2023, doi:10.3390/polym15061414_

Round 1

Reviewer 1 Report

Manuscript Number: polymers-2176601                                       

Full Title: Effects of pine pollen polysaccharides and sulfated polysaccharides on ulcerative colitis and gut flora in mice

In general manuscript describes effects of pine pollen polysaccharides and sulfated polysaccharides on ulcerative colitis and gut flora in mice. The manuscript is well organized, results and discussions are adequate. Hence manuscript may be accepted for publication in the journal of ‘polymer’ in the present form.

Author Response

In general manuscript describes effects of pine pollen polysaccharides and sulfated polysaccharides on ulcerative colitis and gut flora in mice. The manuscript is well organized, results and discussions are adequate. Hence manuscript may be accepted for publication in the journal of ‘polymer’in the present form.

A: Thank you for your comments. We are very grateful to you for agreeing with and supporting our research and conclusions.

Reviewer 2 Report

Section Materials and methods: Would you like to describe more information abut the methods and instruments used in the study (not only a list of them) ?

Please provide a information about the procedure for preparation of SPPM60?

All figures must to be revised.  The text is unread.

Author Response

Section Materials and methods: Would you like to describe more information abut the methods and instruments used in the study (not only a list of them) ?

A: Thank you for your suggestion. We have canceled the list of these information and added them directly to the specific places in ‘Materials and Methods’ section.

Please provide a information about the procedure for preparation of SPPM60?

A: Thank you for your suggestion. We have revised the writing of the ‘Materials and Methods’ section and added the detailed procedure for preparation of SSPM60 in revised 2.3 section.

All figures must to be revised.  The text is unread.

A: Thank you for your suggestion. We have revised all the figures and provided high-quality versions to make the text readable.

Reviewer 3 Report

The chemical structures of pine pollen polysaccharides and sulfated polysaccharides should be added to manuscript.

The chemical structures of polysaccharides must be verified analytical methods.

Is there a structure/activity relationship?

Author Response

The chemical structures of pine pollen polysaccharides and sulfated polysaccharides should be added to manuscript.

A: Thank you for your comment. The focus of this study is to explore the effects of pine pollen polysaccharides (PPM60) and sulfated polysaccharides (SPPM60) on UC mice in vivo. Therefore, we only performed simple physical characterizations of pine pollen polysaccharide and sulfated polysaccharides in this study referring to the published articles[PMCID:PMC6017409], such as monosaccharide composition analysis of PPM60, sulfate substitution degree detection of SPPM60 and infrared spectrum of characteristic absorption peak (details see result 3.1). Regretful, we have not carried out in-depth characterization of the chemical structure in PPM60 and SPPM60. However, we will assay it in future research. 

The chemical structures of polysaccharides must be verified analytical methods.

A: Thank you for your comment. In the present study, we analyzed the monosaccharide composition of polysaccharides using HPLC, revealing that PPM60 was mainly composed of galactose, glucose, xylose, mannose, rhamnose, and an unknown monosaccharide with molar ratio as 12.830:10.449:29.693:1:1.415:1.426. By chlorosulfonic acid-pyridine methods, sulfated polysaccharide was obtained and named SPPM60. The degree of substitution was 1.45 using barium sulfate turbidity method. Finally, the characteristic absorption peaks of PPM60 and SPPM60 were determined using FT-IR spectrophotometer. PPM60 had an absorption peak of O-H, with a stretching vibration at 3372.63 cm-1, and a stretching vibration of C-H appeared at 2929.58 cm-1, which were the characteristic absorption peaks of polysaccharides. SPPM60 had a characteristic absorption peak of S=O at 1224.56 cm-1 and a characteristic absorption peak of C-O-S at 831.45 cm-1. In addition, absorption peak of the O-H stretching vibration of SPPM60 was weaker than PPM60, indicating that -OH was replaced by SO42−.

Is there a structure/activity relationship?

A: Thank you for your comment. Theoretically, many polysaccharides have unique properties and biological activities due to their different structures. The functional diversity is determined by their structural diversity. Sulfated polysaccharides refer to polysaccharides with sulfate radical on sugar hydroxyl, which are the most natural or chemically modified polysaccharides with outstanding effects in the study of structural modification [PMID: 31158418]. After sulfated modification of polysaccharides, the original structure of polysaccharides was changed due to the steric hindrance and electrostatic repulsion effect of sulfated groups, and the water solubility of sulfated polysaccharides was improved, and the activity of polysaccharides was also enhanced to varying degrees. In addition, the activity of sulfated polysaccharides is not only closely related to the existence of sulfate radical, but also related to the degree of sulfate substitution.

Reviewer 4 Report

Congratulations for your study !

Just a small question: do you think the supplementation with  pine pollen polysaccharides and sulfated polysaccharides could affect people having diabetes ?

Author Response

Just a small question: do you think the supplementation with pine pollen polysaccharides and sulfated polysaccharides could affect people having diabetes?

A: Thank you for your comment. According to the current research, various natural polysaccharides have anti diabetes activity, and these polysaccharides are increasingly used in developing countries as adjunctive means of conventional therapy. [for example, PMID: 33687877; PMID: 27083840]. Many studies have shown that Chinese herbal polysaccharide also has good efficacy in the treatment of diabetes [for example, PMID: 30342938; PMID: 32186034]. However, studies on the effects of pine pollen polysaccharides on diabetes have indeed not been reported. This will be a great research direction and we will also conduct further research exploration on the bioactivities of Pinus yunnanensis pollen polysaccharides in diabetes.

Reviewer 5 Report

Abstract R 22 Please check this affirmation  « SPPM60 regulated lipid metabolism-related metabolic pathways, such as glycine, serine, and threonine metabolism »

Pg 2, R41 please mention some of the most important side effects 

Pg 3 R108 How were the animals euthanized and mention the document with the approval of the Ethics Committee 

Why was sulfation of polysaccharides chosen? What evidence exists in this regard? What does it help? 

Pg 4 R132 Please mention a reference for the DAI index 

It is not clearly specified whether the polysaccharides were introduced into the drinking water or administered orally directly. 

Are these polysaccharides and sulfated polysaccharides soluble in water? If not, how did you prepare the suspension and how do you have the certainty that each animal received the mentioned dose? 

How did you choose the dose of 200 mg/kg?

Please specify why you did not use a  group treated with a drug known to be effective in the treatment of ulcerative colitis?

Why did you determine the microbial flora in the cecum, when in ulcerative colitis the changes are present in the colon?

For table 3 and 4, please explain the abbreviations

At the end of the discussions, please make a comparison between the two analyzed products and how these differences would be explained from a structural perspective

Author Response

Abstract R 22 Please check this affirmation  « SPPM60 regulated lipid metabolism-related metabolic pathways, such as glycine, serine, and threonine metabolism »

A: Thank you for your comment. We have checked and revised relevant statements in the Abstract section. The modified expression is ‘On the serum metabolism level, PPM60 and SPPM60 mainly regulated the abnormal serum metabolism of UC mice by regulating the energy-related and lipid-related metabolism pathways, respectively’.

Pg 2, R41 please mention some of the most important side effects 

A: Thank you for your comment. We have listed some of the most important side effects in the corresponding places (see sixth sentence of first paragraph in introduction section).

Pg 3 R108 How were the animals euthanized and mention the document with the approval of the Ethics Committee 

A: Thank you for your comment. The mice were euthanized by cervical dislocation after anesthesia. This animal study was conducted in accordance with the ethical guidelines approved by the Committee for the protection and use of Shandong Normal University Animals (No. AEECSDNU2019042). The detailed description was also supplemented in ‘Materials and Methods’ 2.5.

Why was sulfation of polysaccharides chosen? What evidence exists in this regard? What does it help? 

A: Thank you for your comment. Reports have suggested that after artificially sulfated modification, sulfated natural polysaccharides usually exhibit higher biological activities, such as antiviral, antioxidant, immunomodulatory, and anti-aging properties. And we have added reference to prove this view in four paragraph of ‘introduction’ section. So we choose sulfation of polysaccharides in this study. Hence, we studied the effects of polysaccharides (PPM60) and sulfated polysaccharides (SPPM60), and compared their effects to verify whether sulfated enhance the biological activities of PPM60.

Pg 4 R132 Please mention a reference for the DAI index. 

A: Thank you for your comment. We have added a reference for DAI index in ‘Materials and Methods’ 2.6.

It is not clearly specified whether the polysaccharides were introduced into the drinking water or administered orally directly. 

A: Thank you for your comment. In this study, polysaccharides (PPM60) and sulfated polysaccharides (SPPM60) was dissolved in drinking water and administered to mice at a dose of 200 mg/kg by i.g. We also added the details in in ‘Materials and Methods’ 2.5 and Table 1.

Are these polysaccharides and sulfated polysaccharides soluble in water? If not, how did you prepare the suspension and how do you have the certainty that each animal received the mentioned dose? 

A: Thank you for your comment. Polysaccharides (PPM60) and sulfated polysaccharides (SPPM60) have good water solubility and they can be dissolved in water.

How did you choose the dose of 200 mg/kg?

A: Thank you for your comment. The dosage of PPM60 and SPPM60 was determined through pre-experiment. According to the reference of relevant research, the gradient concentration of 0-400 mg/kg was set in the pre-experiment. Finally, we choose 200 mg/kg because of its small dosage and good effects.

Please specify why you did not use a group treated with a drug known to be effective in the treatment of ulcerative colitis?

A: Thank you for your comment. we used a prevention program. That is, the UC mice model was induced and simultaneously the mice was treated by PPM60 and SPPM60 to evaluate the suppression effect of PPM60 and SPPM60 on the UC occurrence and development in mice. Drugs known to be effective in the treatment of ulcerative colitis are usually given to patients after the disease diagnosis. In this study, PPM60 and SPPM60 intervention was used at the modeling stage. So there is no group treated with a drug known to be effective in the treatment of ulcerative colitis.

Why did you determine the microbial flora in the cecum, when in ulcerative colitis the changes are present in the colon?

A: Thank you for your comment. The intestinal flora mainly exists in the small and large intestine, and the large intestine is the main one. The colon is the primary flora location and workplace. Whereas the cecum resembles a depot of bacteroidetes and balances the flora. Hence, the microbial flora of both cecum and colon were assayed to more fully analyze the effects of PPM60 and SPPM60 on microbial flora in UC.

For table 3 and 4, please explain the abbreviations

A: Thank you for your comment. We have explained the abbreviations of table 2-3 in Results 3.4.2 of  the manuscript. Specially, s means singlet; d means doublet; t means triple peak; q means quartet; dd means doublet doublet; m means multiplet.

At the end of the discussions, please make a comparison between the two analyzed products and how these differences would be explained from a structural perspective

A: Thank you for your comment. We have made a comparison between Polysaccharides (PPM60) and sulfated polysaccharides (SPPM60) in ‘Conclusions’ section. Theoretically, many polysaccharides have unique properties and biological activities due to their different structures. The functional diversity is determined by their structural diversity. Sulfated polysaccharides refer to polysaccharides with sulfate radical on sugar hydroxyl, which are the most natural or chemically modified polysaccharides with outstanding effects in the study of structural modification [PMID: 31158418]. After sulfated modification of polysaccharides, the original structure of polysaccharides was changed due to the steric hindrance and electrostatic repulsion effect of sulfated groups, and the water solubility of sulfated polysaccharides was improved, and the activity of polysaccharides was also enhanced to varying degrees. In addition, the activity of sulfated polysaccharides is not only closely related to the existence of sulfate radical, but also related to the degree of sulfate substitution.

Round 2

Reviewer 2 Report

My opinion i that the manuscript in successfully revised. However, the quality of the  Figures (fig. 4 and 6 ) is still unreadable.

Author Response

 A: Thank you for your suggestion. We have revised the quality of figure 4 and figure 6 (see revised fig.4 and revised fig.6). In addition, we have checked and revised the error of English language and writing in the manuscript to make the text more readable.

Reviewer 5 Report

You stated that you used a prophylactic treatment, not a curative one. Please keep this idea in the text.

Please explain the abbreviation i.g.

Author Response

You stated that you used a prophylactic treatment, not a curative one. Please keep this idea in the text.

A: Thank you for your comment. According you suggestion, we added the detail of this idea in ‘Materials and Methods’ 2.5 section.

Please explain the abbreviation i.g.

A: Thank you for your comment. The abbreviation i.g means intragastrical administration. We also added the explain in ‘Materials and Methods’ 2.5 section and Table 1.